# How sex impacted associations between psychological distress and worry on adults' health behaviours during SARS-CoV-2

Alysha L. Deslippe[1,2], Kim L. Lavoie[3,4], Simon L. Bacon[4,5], Tamara R. Cohen[1,2]*, on behalf of the iCare Platform[4]

1 Faculty of Land and Food Systems, University of British Columbia, Vancouver, British Columbia, Canada, 2 BC Children's Hospital Research Institute, Healthy Starts, Vancouver, British Columbia, Canada, 3 Psychology, Université du Québec à Montréal, Montreal, Québec, Canada, 4 Montreal Behavioural Medicine Centre, CIUSSS-NIM – Hopital du Sacre-Coeur de Montreal, Montreal, Québec, Canada, 5 Health, Kinesiology and Applied Physiology, Concordia University, Montreal, Québec, Canada

* tamara.cohen@ubc.ca

## Abstract

Severe acute respiratory syndrome coronavirus 2 (SARS-CoV-2) has been associated with poorer mental well-being (e.g., anxiety, depressive symptoms and infection worry) and unfavorable changes in health behaviours (e.g., physical activity, healthy eating, and alcohol intake). Notably, trends in changes appear to differ between adult males and females (>18 years). To investigate these sex-based differences, we explored the impact of sex as a moderator on changes in health behaviours attributed to changes in mental well-being during SARS-CoV-2. Data from Canada, Columbia and Ireland collected through the International COVID-19 Awareness and Responses Evaluation (iCARE) Study was used. Participants (n = 17,880; 52.3% female) self-reported changes in their mental well-being and health behaviours attributed to SARS-CoV-2. Associations were explored using multi-variable logistic regression models stratified by country. Significance was determined by p < .01. In Canada and Ireland, increased psychological distress was significantly associated with ≥50% increase in the odds of an unfavourable change in healthy eating, physical activity and alcohol intake (minimum odds ratios (ORs)=1.50). In Columbia, increased psychological distress was significantly associated with unfavourable changes in only alcohol intake (OR=1.63). Increased infection worry was associated with unfavourable changes in physical activity (OR=1.21) in Columbia. Sex was found to significantly moderator associations in Columbia; Females experienced more unfavourable changes in their odds of healthy eating compared to males with increased psychological distress (p < .01). Given the lack of consistent sex-based trends across countries, geographic and cultural context are likely more salient to tailored future behavioural interventions intended to support adults' practice of health-protective behaviours during a pandemic than biological sex (e.g., public health campaign promoting

**Data availability statement:** This is secondary analysis of cross-sectional data collected as a part of the International COVID-19 Awareness and Responses Evaluation (iCARE) Study led by the Montreal Behavioural Medicine Centre (MBMC; a joint Centre intégré universitaire de santé et de services sociaux du Nord-de-l'Île-de-Montréal (CIUSSS- NIM)/Université du Québec à Montréal/Concordia University academic research and training centre). All data available by request from the www.icarestudy.com website.

**Funding:** This iCARE study was supported by the Canadian Institutes of Health Research (CIHR: MM1-174903; MS3-173099; SMC-151518); the Medical Science Fonds of the Major of Vienna (COVID020); the Canada Research Chairs Program (950- 232522, Chair holder: Dr. Kim L. Lavoie); the Fonds de recherche du Québec—santé (FRQ-S: 251618; 34757); the Fonds de recherche du Québec société et culture (FRQSC: 2019-SE1-252541); and the Ministère de l' Économie et de l'Innovation du Québec (2020-2022-COVID-19-PSOv2a- 51754). Study sponsors had no role in conducting the research. None of the funders were involved in the study design or activities of this manuscript.

**Competing interests:** Two authors (KL and SB) were included in the set-up of iCARE and management its current activities. All other authors have no financial or conflicts of interest to declare.

culturally relevant physical activity in response to greater perceived psychological distress).

## Introduction

Severe acute respiratory syndrome coronavirus 2 (SARS-CoV-2) is an infectious disease that was declared an international pandemic by the World Health Organization in March 2020 [1]. The disease had widespread impact across the globe. Common physical symptoms of the disease include respiratory distress, and in extreme cases death [2]. Global disruptions in food [3], social [4] and economic systems were also observed [4,5]. These disruptions in combination with country-specific approaches to manage SARS-CoV-2 (e.g., lockdowns, curfews and distancing measures) led to substantial changes in practiced behaviours [6,7].

SARS-CoV-2 led to many unfavourable impacts on health behaviours that were observed worldwide, including changes to physical activity and dietary behaviours [8–12]. During the first year of the pandemic, global reductions in the intensity [8] of physical activity was reported in international samples. In France [9] and Zimbabwe [10], reductions in the frequency of physical activity was also found. When it came to dietary behaviours, increases in the frequency of eating [8,11] and pursuit of seeking out comfort foods that are typically higher in fat, salt and sugar [11] were reported. Cross-sectional studies conducted in Zimbabwe [10] and Canada [13] also found that alcohol consumption increased. These changes are problematic as reduced physical activity, increased consumption of certain nutrients (e.g., fat, sugar and salt) and greater alcohol intake are known risk factors for poorer health outcomes [14].

Similar to changes in health behaviours, global data reported poorer mental well-being among adults [12,13,15–22]. This included increased reports of anxiety and depressive symptoms in Canada [12,13], China [20,21], Israel [19], Italy [15], the United States [16,22], and international samples [17,18]. In addition, increased worry [22] and fear were reported in Italy [15], the United States [16,22] and international samples [17]. Although these changes are multi-factorial, they likely contributed to the deterioration of health behaviours during SARS-CoV-2. Greater symptoms of anxiety and depression have been linked to impaired or poorer health behaviours and represent greater psychological distress in literature from Canada [23] and the United States [24,25]. In the context of SARS-CoV-2, a scoping review of 23 papers found that higher levels of anxiety and depression were associated with unfavourable dietary habits (e.g., eating more energy-dense foods and drinking more alcohol) and physical activity practices (e.g., less time spent walking and being active) [11]. Perseverative cognition, like worry and fear, that involve continuously thinking about negative events in the past or future have also been shown to impact engagement in health behaviours [26,27]; A meta-analysis of 19 studies found that perseverative cognition was associated with an increased risk of drinking alcohol and eating less healthy [27].

Not all individuals experienced unfavourable shifts in their health behaviours during SARS-CoV-2, nor experienced poorer mental well-being. For example, a

cohort study from France (n = 37,252) found that some adults had favourable changes in their diet and physical activity behaviours compared to their pre-lockdown self-reported behaviours [9]. A critical factor correlated with a greater likelihood of unfavourable changes in health behaviours within the French cohort was found to be sex; females reported more unfavourable changes in their health behaviours in comparison to males [9]. Given that behavioural changes are impacted by diverse factors it is critical to clarify underlying mechanisms of changes to guide the development of future interventions intended to support maintenance of health protective behaviours during a the next pandemic [28].

Sex is rooted in biological mechanisms and is not to be conflated with gender. Gender is a construct rooted in social norms and personal identity [29] that can impact health behaviours through social mechanisms. For example, preferences for leaning on peers as a form of social support during the pandemic has been associated with gender identity and what norms are considered 'acceptable' for different groups to enact [13,30]. In the context of sex, literature has proposed that biological mechanisms may impact behaviour changes during times of great stress, deemed as 'stress coping' [31,32]. Distinctions between sex and gender on associated changes in behaviours are critical to untangle as an intervention targeting a biological mechanism compared to social norms are vastly different.

In the context of SARS-CoV-2, limited cross-sectional literature has found differences in mental well-being changes and behavioural trends based on sex. In Zimbabwe, a study reported worse physical activity behaviours correlated with greater anxiety among females compared to males [10]. In the context of dietary behaviours changes attributed to SARS-CoV-2, two cross-sectional studies from Canada found worse healthy eating behaviours that were associated with greater self-reported stress among females, but not males [12], and significantly higher alcohol intake correlated with increased stress, loneliness and hopelessness among males, but not female [13]. In the United Kingdom, greater report of unfavourable shifts in dietary behaviours during SARS-CoV-2 were observed among males compared to females [33]. Considering the limited number of studies and the inconsistencies they present in sex-based trends, firm conclusions about how males and females have been impacted globally cannot be inferred at this time. To help address this gap, we aim to explore sex as a possible mechanism underlying changes in mental well-being and health behaviours during SARS-CoV-2.

One effective method to explore the impact of sex on behavioural practices during SARS-CoV-2 is through the use of country-level comparisons. These comparisons are effective to this end as trends in sex should transcend geographic and cultural boarders given that they are rooted in biological mechanisms [29]. In contrast, gender-based trends that are often conflated with sex in previous literature are place-based and vary across countries and cultures. As such, we aimed to explore the consistency of trends in the role of sex as a moderator on changes in associations between mental well-being (e.g., psychological distress and worry) and health behaviours (e.g., physical activity, healthy eating, and alcohol intake) across three countries (Canada, Columbia and Ireland) attributed to SARS-CoV-2.

## Materials and methods

This is a secondary analysis of cross-sectional data collected as a part of the International COVID-19 Awareness and Responses Evaluation (iCARE) Study led by the Montreal Behavioural Medicine Centre (MBMC; A joint Centre intégré universitaire de santé et de services sociaux du Nord-de-l'Île-de-Montréal (CIUSSS- NIM)/Université du Québec à Montréal/Concordia University academic research and training centre). Ethics for the iCARE study was approved by the Comité d'Ethique de la Recherche du Centre Intégré Universitaire de Santé et de Services Sociaux du Nord- de- l'Île- de- Montréal (reference number 2020-2099/25-03-2020). All participants in the iCARE study provided informed written consent for their participation and were above the age of 18 years at recruitment. Ethics for access to de-identified data for this manuscript was provided by the University of British Columbia (reference number H23-01045) on 07-13-2023 and the data was received on 08-17-2023 for analyses. Of the 42 countries that participated in the iCARE study, Australia, Canada, Colombia, France, Ireland, Israel, Italy, the United Kingdom and the United States collected data using representative samples of their respective populations based on demographic characteristics. To limit responder bias across countries on any potential confounding demographic factors, we explored data from nationally representative samples in the iCARE

project that were fully collected, cleaned and asked about changes in mental well-being and health behaviours attributed to SARS-CoV-2. This left us with three eligible countries: Canada, Columbia and Ireland.

## iCARE study design

The methods of the iCARE Study (www.icarestudy.com) are described in detail elsewhere [34]. Briefly, participants were recruited from 42 countries to complete electronic surveys about their awareness of local public health responses to SARS-CoV-2, attitudes and beliefs about policies, behaviour changes, concerns, sources of information, viral testing, infection, impacts on school/education, vaccine-related aspects, and sociodemographic factors. A sub-group of participants in the iCARE Study were recruited in waves to form nationally representative samples based on age, sex, region, education level, and income. Each wave was collected for one month at a time and the frequency of waves collected varied depending on the country and local resources. Participants were recruited by email using local polling services and were compensated for their time according to the respective policies of an individual polling service.

Participants were emailed a unique survey link that could only be accessed once. The surveys consisted of 75 questions displayed independently that took an average of 15–20 minutes to complete. Questions appeared at random and included adaptive questioning to reduce participant burden. Survey questions are available online (https://osf.io/nswcm) and were completed in the local language of a country. All responses were automatically stored on secure servers at the MBMC as they were completed. The Checklist for Reporting Results of Internet E-Surveys (CHERRIES) [35] can be found in the supplementary material (S1 Table).

## Eligibility for this analysis

To be included in this analysis, participants from Canada, Columbia and Ireland who completed an online survey from April 2020-December 2022 and reported changes in their health behaviours "since the start of COVID-19 (referring to SARS-CoV-2)" were assessed. Using the CHERRIES definition for completeness [35] 98.0% of responders in Canada, 95.5% in Columbia and 78.7% in Ireland completed the survey from eligible waves. A full description of which participants were included can be found in Fig 1.

## Measures

*Psychological distress* was assessed as a summary variable using two items: 1. Changes in depressive symptoms; and 2. Changes in anxiety symptoms. Perceived changes in feelings of depression were measured by a single item asking participants to 'rate the extent to which SARS-CoV-2 has impacted the following aspects of your life over the last month: because of COVID-19 (referring to SARS-CoV-2) I have felt sad, depressed, or hopeless' with 5-point Likert style responses of 'not at all', 'very little', 'somewhat' and 'to a great extent' possible. Perceived increases in feelings of anxiety were asked in the same way (i.e., I have felt nervous, anxious, or worried). A continuous average was used for the analyses as the two items were highly correlated across all three countries ranging from .73 to .78.

*Infection worry* was created as a summary variable using four items. These items asked participants to self-report their concerns attributed to COVID-19 (referring to SARS-CoV-2) of 'being infected myself,' 'infecting other people I live with,' infecting other people in the community,' and 'the health care system becoming overloaded.' An average of the four items was used for the analyses in Canada ($\alpha$ = .86), Columbia ($\alpha$ = .81) and Ireland ($\alpha$ = .87) as all summary variables displayed moderate (<.80) internal consistency.

*Sex* (male, female) was asked through one item and kept dichotomized for the analysis.

Changes in *health behaviours* were measured using three items. Participants were asked to self-report 'In general, how have the following behaviours changed since the start of COVID-19 (referring to SARS-CoV-2)?' for 'doing physical activity,' 'eating a healthy diet' and 'drinking alcohol.' A six-point Likert-style response was possible with choices of 'I do this a lot more,' 'I do this more,' 'I do this as much as before,' 'I do this less,' 'I do this a little less.' Participants could also select

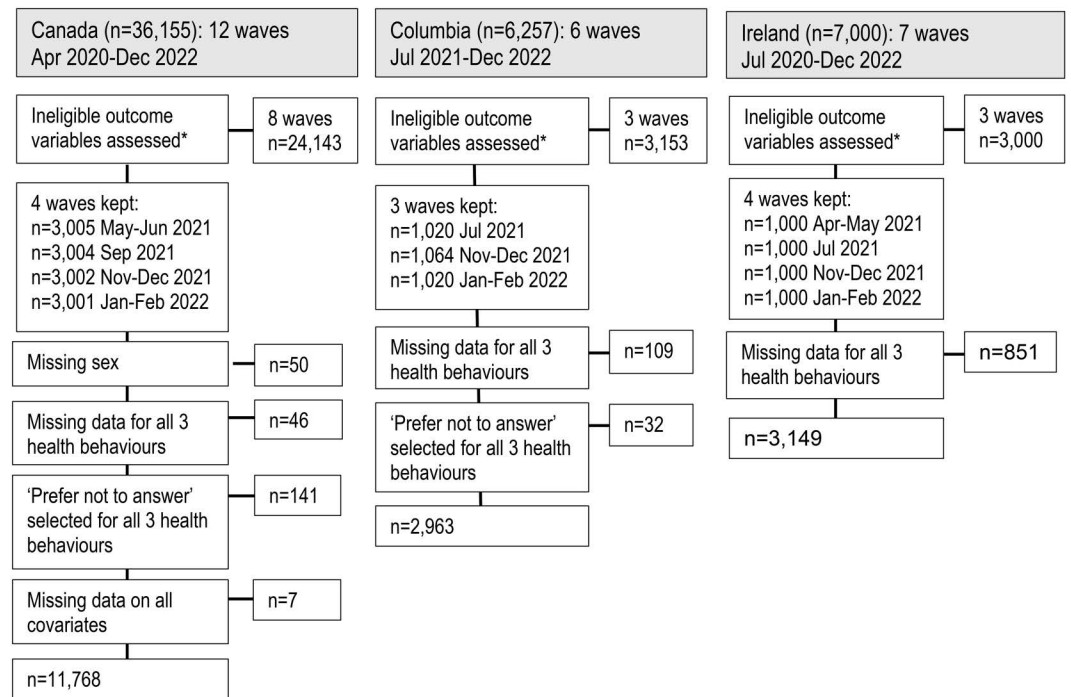

**Fig 1. Flow diagram of eligibility in present analysis from the iCARE study.** Waves from Canada that were include in this analysis are 6, 7, 8 and 9. Waves from Columbia included in the analysis are 1,2 and 3. Waves from Ireland included in the analysis are 2, 3, 4 and 5. *Waves excluded did not assess changes in health behaviours "since the start of COVID-19 (referring to SARS-CoV-2)," and instead focused on other measures of health behaviours.

'I don't do this' or 'I prefer not to answer' for each item. The first two categories were collapsed into one (i.e., increased). The same technique was used for 'I do this less' and 'I do this a little less' (i.e., decreased). Responses of 'I prefer not to answer' were considered missing and excluded from the analysis as the answer was intended to allow participants to skip a question without leaving it fully blank. For analysis, categories were dichotomized into 'unfavorable changes' (i.e., decreased physical activity, decreased healthy eating, and increased alcohol intake) and 'favorable or no changes' (i.e., increased physical activity, increased healthy eating or decreased alcohol intake). This approach was taken as the proportional odds assumption was violated and some groups had small cells (<5%). Responses of 'I don't do this' were excluded from the analyses as these responses do not address our research question to explore changes in each health behaviour if an individual did not partake in that behaviour at all.

Covariates of age (continuous) and having a history of a clinical diagnosis of depression and/or anxiety were also asked (have you ever 'received a clinical diagnosis of…').

## Data analysis

Statistical analyses were conducted in StataSE Version 17.0 [36]. To explore trends in the associations between mental well-being (e.g., psychological distress and worry) and health behaviours (e.g., physical activity, healthy eating, and alcohol intake) we examined: 1. Associations between mental well-being and health behaviour changes (t-tests); 2. Sex and health behaviour changes (chi-square tests); and 3. Sex as a moderator of the associations between mental well-being measures and health behaviour changes (logistic regression model building). All analysis were stratified by country and controlled for age, clinical anxiety, and clinical depression.

To build the logistic regression models and probe for sex as a moderator we followed a two-step process [37]. First, interaction terms between each measure of mental well-being (i.e., psychological distress and infection worry) and sex on associated changes in health behaviours were explored independently from one another. An interaction term with a p-value < .10 was carried into a final model where both psychological distress and worry were explored together [37,38]. In the final model from each country a conservative p-value < .01 was used to indicate statistical significance as we aimed to reduce the likelihood of detecting spontaneous significance given that a high number of models were run.

## Results

Demographic information across all three countries is presented in Table 1. The process of building the final multi-variable models can be found in the supplementary material as S2 Table (Canada), S3 Table (Columbia) and S4 Table (Ireland). Findings from the final multi-variable models for each country are presented below.

### Canada

Females were significantly younger than males (47.2 years vs. 49.8 years), and self-reported more clinical diagnosis of anxiety (25.0% vs. 14.1%) and depression (17.6% vs. 11.8%). Females also reported greater incidence of *psychological*

**Table 1. Descriptive statistics of iCARE study participants from three countries by sex.**

| | Canada n = 11,768 | | | Colombia n = 2,963 | | | Ireland n = 3,149 | | |
|---|---|---|---|---|---|---|---|---|---|
| | Male n = 5,694 | Female n = 6,074 | Sex difference | Male n = 1,222 | Female n = 1,741 | Sex difference | Male n = 1,605 | Female n = 1,544 | Sex difference |
| **Covariates** | | | | | | | | | |
| Age | 49.8 (16.6) | 47.2 (16.8) | 8.5* | 39.7 (14.0) | 35.3 (12.5) | 8.8* | 52.2 (14.1) | 44.2 (14.3) | 15.9* |
| Clinical anxiety | | | 226.2* | | | 9.0* | | | 96.6* |
| *No* | 4,794 (84.2) | 4,419 (72.8) | | 1,050 (85.9) | 1,420 (81.6) | | 1,367 (85.2) | 1,101 (17.3) | |
| *Yes* | 803 (14.1) | 1,519 (25.0) | | 137 (11.2) | 260 (14.9) | | 181 (11.3) | 378 (24.5) | |
| Clinical depression | | | 81.4* | | | 3.1 | | | 30.9* |
| *No* | 4,992 (87.7) | 4,866 (80.1) | | 1,071 (87.6) | 1,493 (85.8) | | 1,406 (87.6) | 1,249 (80.9) | |
| *Yes* | 670 (11.8) | 1,068 (17.6) | | 113 (9.2) | 196 (11.3) | | 134 (8.3) | 225 (14.6) | |
| **Independent variables** | | | | | | | | | |
| Psychological distress | 1.13 (.95) | 1.51 (.97) | −.21.6* | 1.29 (.99) | 1.54 (1.03) | −6.6* | 1.39 (.98) | 1.83 (.93) | −12.6* |
| Infection worry | 2.17 (.87) | 1.99 (.83) | 11.0* | 2.42 (.69) | 2.53 (.63) | −4.6* | 2.09 (.85) | 2.28 (.75) | −6.7* |
| **Dependent variables** | | | | | | | | | |
| Healthy eating | | | 78.8* | | | 16.5* | | | 34.4* |
| *Increase/no change* | 4,560 (80.0) | 4,489 (73.9) | | 855 (70.0) | 1080 (62.0) | | 1,256 (78.3) | 1,057 (68.5) | |
| *Decrease* | 969 (17.0) | 1,440 (23.7) | | 264 (21.6) | 479 (27.5) | | 308 (19.2) | 452 (29.3) | |
| Physical activity | | | 45.6* | | | 16.9* | | | 22.1* |
| *Increase/no change* | 3,610 (63.4) | 3,416 (56.2) | | 751 (61.5) | 901 (51.8) | | 1,129 (70.3) | 962 (62.3) | |
| *Decrease* | 1,857 (32.6) | 2,291 (37.7) | | 400 (32.7) | 667 (38.3) | | 429 (26.7) | 528 (34.2) | |
| Alcohol | | | 2.5 | | | 0.0 | | | 7.88* |
| *Increase/no change* | 3,359 (60.0) | 3,117 (51.3) | | 671 (54.9) | 804 (46.2) | | 967 (60.2) | 778 (50.4) | |
| *Decrease* | 1,002 (17.6) | 1,008 (16.6) | | 214 (17.5) | 255 (14.6) | | 342 (21.3) | 354 (22.9) | |

Number (percent) presented except for independent variables where mean (standard deviation) presented. n = number. Age measured in years. Percentages do not add to 100% in all cases due to missing data and rounding. Independent variables originally measured on a scale of 1–5. The larger a number the worse the perceived change in health behaviour.

* p < .01 calculated using t-tests or chi-squared tests for within country differences between males and females.

distress (1.51 vs. 1.13) but lower *infection worry* than males (1.99 vs. 2.17). When it came to health behaviours, females reported more reduction in healthy eating (23.7% vs. 17.0%) and physical activity (37.7% vs. 32.6%).

*Psychological distress* had a positive association with greater odds of participants reporting reduced healthy eating (OR=1.82) and physical activity (OR=1.50), as well as increased alcohol intake (OR=1.86) (Table 2). *Infection worry* was only associated with increased odds of reduced healthy eating (OR=1.08).

No interaction terms in the multivariable models were significant. Instead, *sex* had significant independent effects with unfavourable changes in healthy eating (OR=1.39) and alcohol intake (OR=.58); Compared to males, females decreased their odds of healthy eating by 39% whereas males increased their odds of alcohol intake by 42% compared to females.

## Columbia

Females were significantly younger than males (35.3 years vs. 39.7 years) and self-reported more clinical diagnosis of anxiety (14.9% vs. 11.2%). Females also reported greater incidence of *psychological distress* (1.54 vs. 1.29) and *infection worry* than males (2.53 vs. 2.42). When it came to health behaviours, females reported greater incidence in their reduction of healthy eating (27.5% vs. 21.6%) and physical activity (38.3% vs. 32.7%).

*Psychological distress* had a positive association with greater odds of participants reporting increased alcohol intake (OR=1.63) only (Table 3). *Infection worry* was only associated with increased odds of reduced physical activity (OR=1.21).

*Sex* significantly moderated the associations between increased psychological distress and reduced healthy eating (p < .01). A one-unit increase in *psychological distress* contributed to a 30% increase in the odds of reduced healthy eating (OR=1.30) in Columbia females. In contrast, a 1% decrease in the odds of reduced healthy eating was reported among Columbian males (OR=.99). No independent effects of *sex* were significant across any health behaviour.

## Ireland

Females were significantly younger than males (44.2 years vs. 52.2 years), and self-reported more clinical diagnosis of anxiety (24.5% vs. 11.3%) and depression (14.6% vs. 8.3%). Females also reported greater incidence of *psychological distress* (1.83 vs. 1.39) and *infection worry* than males (2.28 vs. 2.09). When it came to health behaviours, females

**Table 2. Associations between sex, psychological distress and worry and health behaviours among Canadian adults during SARS-CoV-2.**

| Canada | Healthy eating | Physical activity | Alcohol intake |
|---|---|---|---|
| | n = 10,877 | n = 10,650 | n = 8,090 |
| Psychological distress | 1.82 (1.63–2.02)* | 1.50 (1.40–1.60)* | 1.86 (1.71–2.03)* |
| Infection worry | 1.08 (1.00–1.17)* | .98 (.92–1.05) | 1.00 (.89–1.13) |
| Sex | 1.39 (1.08–1.79)* | 1.09 (.98–1.22) | .58 (.39–.85)* |
| Psychological distress x sex | .91 (.79–1.04) | | |
| Worry x sex | | | 1.16 (.98–1.38) |
| Age | .99 (.98–.99)* | 1.00 (1.00–1.00) | .99 (.98–.99)* |
| Clinical anxiety | 1.11 (.93–1.33) | 1.06 (.90–1.25) | 1.05 (.85–1.29) |
| Clinical depression | 1.34 (1.10–1.62)* | 1.13 (.94–1.34) | 1.22 (.97–1.52) |
| Wave 7 | .82 (.69–.98)* | .85 (.73–.99)* | .87 (.72–1.06) |
| Wave 8 | .89 (.75–1.07) | .90 (.77–1.04) | .91 (.74–1.11) |
| Wave 9 | .91 (.76–1.08) | 1.10 (.95–1.27) | .89 (.73–1.09) |
| Pseudo r ^2 | .07 | .03 | .08 |

Odd ratios and 99% confidence intervals shown. Reference group for sex (male), clinical anxiety (no diagnosis), clinical depression (no diagnosis), wave (6). n = number.

* p < .01.

**Table 3. Associations between sex, psychological distress and infection worry and health behaviours among Columbian adults during SARS-CoV-2.**

| Columbia | Healthy eating | Physical activity | Alcohol intake |
|---|---|---|---|
| | n = 2,549 | n = 2,581 | n = 1,857 |
| Psychological distress | .99 (.81–1.20) | 1.09 (.92–1.29) | 1.63 (1.38–1.91)* |
| Infection worry | 1.45 (.94–1.40) | 1.21 (1.01–1.45)* | .99 (.77–1.26) |
| Sex | .86 (.56–1.31) | 1.02 (.70–1.48) | .83 (.61–1.12) |
| Psychological distress x sex | 1.31 (1.03–1.67)* | 1.78 (.95–1.46) | |
| Age | .99 (.98–1.00)* | 1.01 (1.00–1.01) | .98 (.96–.99)* |
| Clinical anxiety | 1.03 (.68–1.57) | .97 (.65–1.43) | 1.31 (.80–2.14) |
| Clinical depression | 1.08 (.68–1.71) | 1.11 (.72–1.71) | .95 (.55–1.65) |
| Wave 2 | .97 (.73–1.29) | .87 (.67–1.13) | .97 (.65–1.46) |
| Wave 3 | .83 (.63–1.11) | .50 (.38–.65)* | 1.64 (.115–2.34)* |
| Pseudo r ^2 | .02 | .03 | .08 |

Odd ratios and 95% confidence intervals shown. Reference group for sex (male), clinical anxiety (no diagnosis), clinical depression (no diagnosis), wave (1). n = number.

* p < .01.

reported greater incidence in their reduction of healthy eating (29.3% vs. 19.2%), physical activity (34.2% vs. 26.7%) and increased alcohol intake (22.9% vs. 21.3%).

*Psychological distress* had a positive association with greater odds of participants reporting reduced healthy eating (OR=1.84) and physical activity (OR=1.62), as well as increased alcohol intake (OR=1.69) (Table 4).

No interaction terms in the multivariable models were significant, nor was *sex* found to have any significant independent effects.

## Discussion

We explored associations between psychological distress and infection worry with self-reported changes in heath behaviours in three countries. We further assessed whether these associations were moderated by sex. Overall, we found

**Table 4. Associations between sex, psychological distress and infection worry and health behaviours among Irish adults during SARS-CoV-2.**

| Ireland | Healthy eating | Physical activity | Alcohol intake |
|---|---|---|---|
| | n = 2,908 | n = 2,885 | n = 2,320 |
| Psychological distress | 1.84 (1.51–2.24)* | 1.62 (1.42–1.85)* | 1.69 (1.45–1.96)* |
| Infection worry | .91 (.78–1.07) | 1.01 (.87–1.17) | .89 (.75–1.05) |
| Sex | 1.61 (.94–2.75) | 1.05 (.84–1.32) | .92 (.71–1.20) |
| Psychological distress x sex | .84 (.65–1.09) | | |
| Age | .98 (.97–.99)* | .99 (.98–1.00) | .99 (.98–1.00)* |
| Clinical anxiety | 1.54 (1.10–2.16)* | 1.32 (.95–1.82) | 1.22 (.84–1.78) |
| Clinical depression | .79 (.53–1.19) | .85 (.58–1.25) | .88 (.57–1.36) |
| Wave 3 | .82 (.59–1.13) | .96 (.71–1.29) | .96 (.69–1.34) |
| Wave 4 | .96 (.69–1.32) | 1.16 (.86–1.57) | .83 (.59–1.17) |
| Wave 5 | .90 (.64–1.25) | .98 (.72–1.33) | .77 (.54–1.10) |
| Pseudo r ^2 | .08 | .05 | .06 |

Odd ratios and 95% confidence intervals shown. Reference group for sex (male), clinical anxiety (no diagnosis), clinical depression (no diagnosis) and wave (2). n = number.

* p < .01.

that sex only moderated the association between increased psychological distress and healthy eating in Columbia, with females having the more deleterious changes compared to males. This suggests that country specific context, instead of biological sex, may be more salient to tailor for in future behavioural interventions aiming to support adults' maintenance of health behaviours during a pandemic.

### Psychological distress had a consistent role in the deterioration of adults' health behaviours compared to infection worry

Increases in psychological distress were associated with unfavourable changes in all three health behaviours regardless of sex in Canada and Ireland, and increased consumption of alcohol in Columbia. In contrast, infection worry only had significant associations with unfavourable changes in healthy eating in Canada and physical activity in Columbia. These trends may suggest that poorer mental well-being (i.e., feeling anxious and sad) had a more impactful role in altering health behavioural practices opposed to specific distress related to SARS-CoV-2. In light of this finding, public health campaigns during a global pandemic may benefit from addressing general changes in mental well-being in their wording (e.g., if you feel anxious today seek out…) opposed to targeted language related to the pandemic (e.g., if you feel worried about SARS-CoV-2 today seek out…)

Differences in the associations may also be explained by cultural and geographic similarities between Canada and Ireland. Canada and Ireland are both high income countries [39] and this has in part contributed to greater spending and availability of services targeting mental well-being in Canada [40] and Ireland [41] compared to funding and services available in Columbia [42]. It is possible that this spending and availability of services contributed to greater acceptance of altering healthy eating and physical activity as a form of coping with poorer mental well-being and especially psychological distress. The warmer climate within Columbia may have also helped mitigate reduced physical activity in particular compared to Canada and Ireland that occurred with SARS-CoV-2 lockdown orders and closure of athletic spaces like gyms [11]. With lockdown orders, adults in Canada and Ireland may not have had access to their typical indoor spaces that promote physical activity. Given the colder climate, it may not have been as feasible for Canadian and Irish adults to pivot to outdoor physical activities, depending on the time of year, in comparison to Columbia adults.

Irrespective of country, greater psychological distress appears to plays a critical role in the deterioration of adults' health behaviours during SARS-CoV-2. Future public health resources meant to support adults' mental well-being during a pandemic should focus on helping adults identify general changes in their psychological distress, and promote practice of feasible health behaviour alternatives to cope (e.g., switching to outdoor physical activity during a lock down).

### Sex had country specific effect on diverse health behaviours

Given that sex-based trends were not consistently reported across geographic boarders, it is likely that place-based factors, like cultural norms and gender norms, may be more salient factors to guide the development efforts of future behavioural interventions seeking to promote practice of health behaviours among male and female adults during a pandemic.

In Canada, sex had significant independent associations with reduced healthy eating and increased alcohol intake. Specifically, females expressed higher odds of reduced healthy eating, but lower odds of increased alcohol intake. These findings may be explained by traditional coping mechanisms expressed among Canadians. For example, Canadian males have generally report higher alcohol intake than females, and this is in part related to social norms of drinking tendancies [13]. Thus, males in Canada may have utilized alcohol consumption as a coping mechanism more so than females during SARS-CoV-2 attributed to traditional social norms surrounding drinking differences. Our findings contrast data from the United States where greater SARS-CoV-2 psychological distress (e.g., depressive symptoms and negative psychological health) was correlated with increases in the number of drinks females consumed, but not males, in a single drinking occasion at home during the pandemic [43]. This reinforces the notion that country specific factors are likely to play an influential role in behaviour changes.

In Columbia, sex significantly moderated the association between psychological distress and healthy eating with females experiencing more deleterious changes compared to males. Females in Columbia may have been uniquely positioned to alter their eating habits in times of stress compared to those in Canada and Ireland. Literature has shown that females whose gender identity corresponds with being a women and girl may be more susceptible to emotional eating [44–46], and cope with negative emotions by consuming comfort foods that are typically higher in fat, sugar and salt. These trends may not have been observed in Canada and Ireland due to the entrenchment of the Western diet compared to Columbia. A Western diet is already high in fat, sugar and salt in Canada and Ireland and as such, females irrespective of their gender identity may have already consumed an elevated amount of these types of comfort foods [47]. As we did not measure specific dietary behaviours, like pursuit of comfort foods, future research is needed to confirm this hypothesis.

### Females may have greater inherent risk factors for unfavourable changes across countries

In all three countries females were significantly younger than males and self-reported greater incidence of having a clinical diagnosis of anxiety. This is noteworthy as younger age and self-report of a clinical diagnosis of anxiety may indicate greater overall awareness of one's mental well-being [48,49]. As such, it is possible that females across the three countries in this analysis were more aware (and thus likely) to report changes in their mental well-being prior to SARS-CoV-2. However, an international cross-sectional study of 8317 adults (68.7% female) found that age had no impact on the likelihood of following national health mandates (e.g., wearing a mask) [50]. Thus, age may not have a critical impact across all health behaviour changes during a pandemic and future research should continue to clarify when age is impactful, and how this intersects with other behavioural predictors.

It is also worth noting that in Canada (25.0% vs. 14.1%) and Ireland (24.5% vs. 11.3%) females reported almost double the incidence of having a clinical diagnosis of anxiety. In contrast, incidence rates between Columbian females and males were not starkly different (14.9% vs. 11.2%). Further, significant differences between females and males self-report of a clinical diagnosis of depression arose in Canada (17.6% vs. 11.8%) and Ireland (14.6% vs. 8.3%), but not Columbia. Taken together, this may suggest that there could be situational factors, such as income level or cultural norms, that are more similar between Canada and Ireland that are impacting the trends within this analysis. This reinforces the merit of considering country level comparisons in behavioural studies to help parse out when situational factors are at play.

### Limitations and strengths

This study is not without limitations. First, the use of self-report data is subject to response bias. Due to how changes in health protective behaviours were measured, we are also unable to establish causation, positive coping, or changes over time. As it is established that increased physical activity and consumption of a healthy diet can have positive impacts on depressive and anxiety symptoms [51,52], we are limited in assessing this relationship unidirectionally in our analyses. These limitations suggest a need for more objective measurement of changes in mental well-being and health behaviours in future research (e.g., use of medical charts) and longitudinal studies to provide a clearer picture of how SARS-CoV-2 impacted health behaviours. The nature of how sex was assessed in our analyses may have also impacted the findings; Sex was assessed in a single item asking participants to "identify their sex." This approach is more ambiguous as this type of wording is more typical of the assessment of gender identity, and may have confused responders. Finally, we were unable to explore data from all 42 countries in the iCARE Study due to differences in how samples were collected from diverse countries (i.e., nationally representative versus convenience), which questions were asked (i.e., countries could tailor their wording to national health aims) and which countries had their data fully collected and cleaned (i.e., not in progress). It would be incredibly insightful to explore trends across all 42 countries, had it been possible, to consider a more complete narrative of when trends may prevail and why.

Despite these limitations, this study has several strengths. By comparing across three different countries, sex-based trends that are rooted in biological mechanisms can be more clearly singled out. In particular, our use of data from Canada, Columbia and Ireland is noteworthy as a large amount of literature exploring psychology and behaviour changes comes from the United States at this time [50]. We further had a large sample with high completeness rates that contributed to high statistical power in our analyses and given the conservative approach we took to model building, were unlikely to be impacted by spontaneous significance. Finally, exploration of psychological distress alongside infection worry is useful in understanding if different indicators of mental well-being may be more salient to consider during a pandemic, helping guide the focus of future public health campaigns.

## Conclusions

SARS-CoV-2 had drastic impacts on the mental well-being and health behaviours of adults globally. Understanding who may be at the greatest risk of unfavorable changes in health protective behaviours is important to guide the development of future behavioural interventions during the next global pandemic. Sex appeared to have unique associations on behavioural changes across Canada, Columbia, and Ireland. As such, it may be more salient to consider country specific behavioural interventions that account for factors like cultural norms, gender norms and economic status in their content and design. Such interventions may benefit from tailored public health campaigns to promote local awareness of signs of increased psychological distress, and feasible behavioural strategies to navigate them (e.g., resources to help manage weather in Canada and Ireland to preform outdoor physical activity during a lock down). Though the trends in the role of sex were inconsistent, across all three countries females reported greater incidence of having a prior clinical diagnosis of anxiety, worse psychological distress, and negative changes to their physical activity and dietary behaviours. This suggests that sex is still an important factor in the discussion of mental well-being and health behaviour changes and should not be simply ignored.

## Supporting information

**S1 Table. Checklist for Reporting Results of Internet E-Surveys (CHERRIES).** This checklist has been modified from Eysenbach G. Improving the quality of Web surveys: the Checklist for Reporting Results of Internet E-Surveys (CHERRIES). J Med Internet Res. 2004 Sep 29;6(3):e34.
(DOCX)

**S2 Table. Logistic regressions exploring associations between sex, psychological distress and worry on adults' health behaviours in Canada during SARS-CoV-2.** Odd ratios and 99% confidence intervals shown. Reference group for sex (male), clinical anxiety (no diagnosis), clinical depression (no diagnosis), wave (6). n = number. *p < .01. ᵃmoderator p-value < .10. Interaction term kept in multivariable model.
(DOCX)

**S3 Table. Logistic regressions exploring associations between sex, psychological distress and worry on adults' health behaviours in Columbia during SARS-CoV-2.** Odd ratios and 99% confidence intervals shown. Reference group for sex (male), clinical anxiety (no diagnosis), clinical depression (no diagnosis), wave (1). n = number. *p < .01. ᵃmoderator p-value < .10. Interaction term kept in multivariable model.
(DOCX)

**S4 Table. Logistic regressions exploring associations between sex, psychological distress and worry on adults' health behaviours in Ireland during SARS-CoV-2.** Odd ratios and 99% confidence intervals shown. Reference group for sex (male), clinical anxiety (no diagnosis), clinical depression (no diagnosis), wave (2). n = number. *p < .01. ᵃmoderator p-value < .10. Interaction term kept in multivariable model.
(DOCX)

## Acknowledgments

The iCARE Platform consists of students, trainees, volunteers and staff employed by the Montreal Behavioural Medicine Centre that work on research tasks of the international iCARE Platform under the leadership of Dr. Kim Lavoie and Dr. Simon Bacon.

## Author contributions

**Conceptualization:** Alysha L. Deslippe, Kim L. Lavoie, Simon Bacon, Tamara R. Cohen.

**Data curation:** Alysha L. Deslippe.

**Formal analysis:** Alysha L. Deslippe.

**Funding acquisition:** Kim L. Lavoie, Simon Bacon.

**Methodology:** Alysha L. Deslippe, Tamara R. Cohen.

**Resources:** Tamara R. Cohen.

**Software:** Alysha L. Deslippe.

**Supervision:** Tamara R. Cohen.

**Validation:** Kim L. Lavoie, Simon Bacon, Tamara R. Cohen.

**Visualization:** Alysha L. Deslippe.

**Writing – original draft:** Alysha L. Deslippe.

**Writing – review & editing:** Alysha L. Deslippe, Kim L. Lavoie, Simon Bacon, Tamara R. Cohen.

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
