## [Decision Letter · Decision Letter 0]

28 Aug 2025

Dear Dr. Cohen,

Thank you for submitting your manuscript to PLOS ONE. After careful consideration, we feel that it has merit but does not fully meet PLOS ONE’s publication criteria as it currently stands. Therefore, we invite you to submit a revised version of the manuscript that addresses the points raised during the review process.

We look forward to receiving your revised manuscript.

Kind regards,

Prof. Anat Gesser-Edelsburg, Ph.D.

Academic Editor

PLOS ONE

Journal Requirements:

This iCARE study was supported by the Canadian Institutes of Health Research (CIHR: MM1-174903; MS3-173099; SMC- 151518); the Medical Science Fonds of the Major of Vienna (COVID020); the Canada Research Chairs Program (950- 232522, Chair holder: Dr. Kim L. Lavoie); the Fonds de recherche du Québec—santé (FRQ-S: 251618; 34757); the Fonds de recherche du Québec société et culture (FRQSC: 2019-SE1-252541); and the Ministère de l’ Économie et de l’Innovation du Québec (2020-2022-COVID-19-PSOv2a- 51754). Study sponsors had no role in conducting the research. None of the funders were involved in the study design or activities of this manuscript.

3. Thank you for stating the following in the Competing Interests/Financial Disclosure section:

Two authors (KL and SB) were included in the set-up of iCARE and management its current activities. All other authors have no financial or conflicts of interest to declare.

We note that one or more of the authors are employed by a commercial company: iCARE

4. One of the noted authors is a group or consortium The iCARE Study team. In addition to naming the author group, please list the individual authors and affiliations within this group in the acknowledgments section of your manuscript. Please also indicate clearly a lead author for this group along with a contact email address.

5. In the online submission form, you indicated that this is secondary analysis of cross-sectional data collected as a part of the International COVID-19 Awareness and Responses Evaluation (iCARE) Study led by the Montreal Behavioural Medicine Centre (MBMC; a joint Centre intégré universitaire de santé et de services sociaux du Nord-de-l’Île-de-Montréal (CIUSSS- NIM)/Université du Québec à Montréal/Concordia University academic research and training centre). All data available by request from the www.icarestudy.com website.

6. Please remove all personal information, ensure that the data shared are in accordance with participant consent, and re-upload a fully anonymized data set.

Additional guidance on preparing raw data for publication can be found in our Data Policy (https://journals.plos.org/plosone/s/data-availability#loc-human-research-participant-data-and-other-sensitive-data) and in the following article: http://www.bmj.com/content/340/bmj.c181.long .

Reviewers' comments:

Reviewer's Responses to Questions

**Comments to the Author**

1. Is the manuscript technically sound, and do the data support the conclusions?

Reviewer #1: Yes

Reviewer #2: Partly

Reviewer #3: Partly

2. Has the statistical analysis been performed appropriately and rigorously?

Reviewer #1: Yes

Reviewer #2: Yes

Reviewer #3: No

3. Have the authors made all data underlying the findings in their manuscript fully available?

Reviewer #1: Yes

Reviewer #2: Yes

Reviewer #3: Yes

4. Is the manuscript presented in an intelligible fashion and written in standard English?

Reviewer #1: Yes

Reviewer #2: Yes

Reviewer #3: Yes

Reviewer #1: Thank you very much for the opportunity to let me read this manuscript. I would like to address some issues for improving.

1In the abstract , you should provide full nme of COVID19

2. Please provide full named of OR

3. you need to give an example of the tailored program.

4. I just wonder, why you used P<.1 to put variables into the quation and <0.1 for the significant level, normally we used 25%.

Reviewer #2: Dear Authors,

Thank you for your contribution to this important article. Below, I have outlined some comments and suggestions for your consideration:

1. Title

Please consider revising the title to make it more concise.

2. Abstract

The abstract effectively sets the context and highlights key findings of the study, but a few improvements can enhance clarity including:

a) Consider making the abstract short and precise as it is written very long

b) Clarify the implications of sex as a moderator in the results

c) ensure consistent terminology and keep statements concise for better readability

3. Introduction

The introduction effectively outlines the impact of COVID-19. However, consider clarifying some points for better flow and coherence. For instance,

a) provide clearer transitions to enhance readability when mentioning studies

b) ensure that the rationale for the country-level comparison is clearly articulated, emphasizing its relevance to the study's objectives.

4. Methods

a) specify the criteria for compensation to ensure transparency when describing participant recruitment

b) The eligibility criteria could be simplified for better readability

c) clarify how the internal consistency of the infection worry variable was assessed

a) How are the statistical analysis methods clearly linked to the research questions?

5. Results

The results section effectively presents key findings across the three countries, highlighting significant differences in health behaviors related to psychological distress and infection worry. However, consider providing clearer summaries of the statistical findings for each country to enhance readability. For instance, clarify the impact of demographic factors, such as age and clinical diagnoses, on the results to contextualize the findings better. It may also be helpful to include a brief discussion on the significance of the observed sex differences in health behaviors to tie the results back to the study's main objectives.

6. Discussion

a) consider enhancing the clarity of the arguments by summarizing key findings more succinctly.

b) Additionally, while the limitations are well-articulated, it may strengthen the section to emphasize the implications of self-report bias and the need for future longitudinal studies.

7. Conclusion

The conclusions effectively summarize the study's findings and emphasize the significant impact of COVID-19 on mental well-being and health behaviors.

a) consider refining the wording for greater clarity, particularly in the call for country-specific interventions. It may be beneficial to explicitly state the implications of the findings for public health policy in each country.

b) Additionally, while the focus on reducing psychological distress is important, expanding on potential strategies for implementing these interventions could enhance the conclusion's practical relevance.

c) Lastly, reiterating the importance of understanding sex differences in health behaviors could further strengthen the argument for targeted approaches in future pandemics.

Reviewer #3: Title: Associations between psychological distress and infection worry with health behaviours among adults during COVID-19 in three countries: A sex-based analysis

MS ID: PONE-D-25-24290

General Comments:

The manuscript, as stated by authors, uses an exploration approach. In general, this exploration focused on associations between psychological distress (as related to COVID-19 pandemic) and a number of health-related behaviours, while considering the influence of participant sex as a moderator. Additionally, authors selected three nations (Columbia, Canada, and Ireland) for data retrieval. Results, using three separate logistic-regression statistical analyses for each nation, showed varied findings across main variables and sex as a moderator.

COMMENT and RECOMMENDATIONS:

I have many issues concerning the (a) write-up of the manuscript, and (b) the method / results employed by the authors. Given the database the authors are drawing on, I hope the comments and examples given below can further the inherent value of this science and author’s revision of the manuscript.

Line 51: Your first sentence - “Coronavirus disease or COVID-19 is an infectious disease…” is filled with a possible redundancy, ambiguity and inaccuracy. When using “Coronavirus disease” which one are you refereeing to? In terms of the recent pandemic the viral infection is referred to as SARS-CoV-2. The “COVID-19” moniker refers to the pandemic event; this is a difference between the event and the infection. Please write a direct sentence, avoid the use of the little word “or”; the sentence reads like a multiple-choice question; either coronavirus disease or COVID-19 is an infection disease. Please be direct, precise and accurate throughout the manuscript.

Line 66: The authors said, “From a psychological standpoint” anxiety symptom “or” depression are linked to impaired “or” poorer health. Again, please reconsider your literature review write up, and take care in the use of “or”. Using psychology as a “standpoint” appears too general; there are many psychological approaches and fields, - you should be more precise; direct the reader’s understanding to a specific approach or field of psychology concerning anxiety and health.

Line 84: Authors write “Sex is rooted in biological mechanisms, and in the context of COVID-19, appears to influence trends between psychological distress….” Given my read of the overall manuscript, the importance of this paragraph is critical to the logic, analysis, and conclusions of this research. Using biological mechanism as a pretext for understanding sex-differences in the context of COVID-19 is a misapplied concept. Yes, sex determination as being a biological mechanism (e.g., genetic and epigenetic processes) I have no doubts.

But the authors have not included a literature review where sex-differences are associated with socialization, gender development, stereotyping, culturalization that differentiate male/female preferences, expectations, and behaviors. Starting on Line 86, with a discussion about Canada, France, United Kingdom and Zimbabwe, are general descriptions reporting how different sexes have responded differently and have responded differently across differently countries. It is difficult to understand the logic of the authors how the biological mechanism alone (and not a social mechanism) becomes the explanation (i.e., sex as an important moderator – as claimed in the manuscript).

There is no review of sex differences rooted in a social/societal mechanism (e.g., see Social Psychology literature) in the introduction. Because the authors have conceptually focused on the sex-based factor in the write up and analyses, I strongly recommend the authors more thoroughly review and discuss sex-difference literature, especially concerning how sex-differences are understood socially and psychologically (i.e., sex-based preferences and expectations). As is, the authors have dedicated one paragraph to what is a central theoretical explanation to their results and their write up. Please make note of the sub-title of the manuscript, “…: A Sex-Based Analysis”.

Line 97: Authors begin a review of “country-level comparisons” and claim country-level comparisons “are effective method to identify consistent patterns in how sex may influence the associations between …” The authors go on to claim “This is because sex is based on biological factors that transcend geographical or cultural boarders”. Is there not a subtle contradiction implied here between these two contiguous sentences? Country-level comparisons are important to identify how sex influences, but then sex identified as a biological factor transcends the country-level factor? I do note the word difference and word substitution.

Never-the-less, there is little literature review regarding the importance of cultural / national influences on people’s habits and health practices. The authors again appear to neglect the psychological, sociological and cultural approaches to better understand “consistent” and inconsistent patterns among people. Again, I recommend authors to seek the literature, maybe starting in general with Harry Triandis’s psychological approach and what he called Cultural Syndromes. Yes, teasing out theoretically socially rooted sex-differences, and the connection with cultural differences is a challenge. But once again, this manuscript has focused on “country-level comparisons” and has dedicated very little literature review in the introduction on this topic.

Line 106: Authors describe the method in general as a secondary analysis of cross-sectional data. This data is a part of a greater data-base, specified as iCARE and led by MBMC. On Line 118, the authors briefly describe the data base as encompassing 42 countries. The authors then describe the different waves of recruitment. And then on Lines 128-129 the authors identify three nations to be analyzed for this manuscript: Canada, Columbia, and Ireland. I recommend that the authors explain the procedure for how these three nations were selected from the 42 counties listed in the iCARE database. No mention of this potential selection bias was mentioned – nor discussed in the limitation section of the Discussion. I refer the authors to the Clark et al. reference (see below) and take special note of the section 2.11 on p. 77. This section might be a good model for writing up a revision of this procedure.

Line 187: Section on Data Analysis describes the “exploration” of data by first seeking difference ( “t-test and chi-square tests” ) between selected countries (i.e., Canada, Columbia, and Ireland). The authors describe using logistic regression as means to isolate significant relations between variables for each country separately. Thus, I assume three different analyses, that is three different logistic regressions (Tables S2, S3, & S4). I realize the authors have identified the methodology as exploratory, but I question why the authors did not build and test an omnibus statistic or model to incorporate a country-level factor with the inclusion of the sex factor? I believe the paper’s logic and strength of contribution becomes compromised throughout by not clearly providing and structuring this overall analysis.

Line 285: A typo in the main heading – “Psychological distress had a consist role in …” – the word “consist” should read “consistent”.

One of the main goals of science is to identify and test variables for the purpose of (a) prediction and (b) treatment/intervention. The many nuanced findings in this manuscript (including similarities and differences) between country-level comparison makes for a difficult interpretation of the variables tested. See Line 337, the authors provide their take on the paper’s strengths. Basically, the strength of the paper is - you can “do” this research. Yes, you can “do” this research – but what did this paper actually tell us about the sex-based factor and the country-level factors? The authors statement - “sex-based trends can be assessed regardless of country-level factors” seems to be an overstatement without evidence provided by the author’s research. Actually, many of the statements given in the manuscript provide evidence concerning the importance of the country-level factor.

For instance, reading over the papers many diverse and singular findings, simply stated here: (a) you show how Canada and Ireland show consistent relations, while Columbia does not, or (b) Canada and Columbia show consistent relations, but not Ireland. Of course, adding sex-factor moderator increases the interpretative nuance. But as an applied health care expert, how do I “use” this information provided in this manuscript? I believe a revision of this manuscript is necessary to clarify the logic, statistical analysis, and the contribution of the findings; that is, how do these variables predict (or do not predict) and how do we then treat the negative effects of a future pandemic (is it at the sex-based factor, country/cultural factor, regime-based factor… etc).

Clark C, Davila A, Regis M, Kraus S. Predictors of COVID-19 voluntary compliance behaviors: An international investigation. Glob Transit. 2020;2:76-82. doi: 10.1016/j.glt.2020.06.003. Epub 2020 Jun 26. PMID: 32835202; PMCID: PMC7318969.

(See p 77, Section 2.11 – the Open Science Statement)

Also, these references might be helpful toward revising the logic and method of this manuscript.

Hensel L, Witte M, Caria AS, Fetzer T, Fiorin S, Götz FM, Gomez M, Haushofer J, Ivchenko A, Kraft-Todd G, Reutskaja E, Roth C, Yoeli E, Jachimowicz JM. Global Behaviors, Perceptions, and the Emergence of Social Norms at the Onset of the COVID-19 Pandemic. J Econ Behav Organ. 2022 Jan;193:473-496. doi: 10.1016/j.jebo.2021.11.015. Epub 2021 Nov 19. PMID: 34955573; PMCID: PMC8684329.

He, Z., Jiang, Y., Chakraborti, R., & Berry, T. D. (2022). The Impact of National Culture on Covid-19 Pandemic Outcomes. Internatinal Journal of Social Economics., 49(3), 313-335 https://doi.org/10.1108/IJSE-07-2021-0424

**Do you want your identity to be public for this peer review?** For information about this choice, including consent withdrawal, please see our Privacy Policy

Reviewer #1: **Yes: ** Nitikorn Phoosuwan

Reviewer #2: No

Reviewer #3: No

---

## [Author Response · Author response to Decision Letter 1]

5 Nov 2025

We thank the reviewers and editor for their time and thoughtful feedback when reviewing this manuscript. Below we have outlined all queries and your responses in italics. Unmarked and marked up versions of the revised manuscript have been attached as separate documents in the online portal. Line numbers referenced below correspond to the marked up version in the portal (not the unmarked version).

Journal feedback:

1. Please ensure that your manuscript meets PLOS ONE's style requirements, including those for file naming. The PLOS ONE style templates can be found at:

We have reviewed the formatting documents and changed the file naming as indicated.

This iCARE study was supported by the Canadian Institutes of Health Research (CIHR: MM1-174903; MS3-173099; SMC- 151518); the Medical Science Fonds of the Major of Vienna (COVID020); the Canada Research Chairs Program (950- 232522, Chair holder: Dr. Kim L. Lavoie); the Fonds de recherche du Québec—santé (FRQ-S: 251618; 34757); the Fonds de recherche du Québec société et culture (FRQSC: 2019-SE1-252541); and the Ministère de l’ Économie et de l’Innovation du Québec (2020-2022-COVID-19-PSOv2a- 51754). Study sponsors had no role in conducting the research. None of the funders were involved in the study design or activities of this manuscript. Please state what role the funders took in the study. If the funders had no role, please state: "The funders had no role in study design, data collection and analysis, decision to publish, or preparation of the manuscript."

The funders had no role in the study design, data collection and analysis, decision to publish or preparation of the manuscript. We have amended the text to include this statement in the cover letter.

3. Thank you for stating the following in the Competing Interests/Financial Disclosure section:

Two authors (KL and SB) were included in the set-up of iCARE and management its current activities. All other authors have no financial or conflicts of interest to declare. We note that one or more of the authors are employed by a commercial company: iCARE. Please provide an amended Funding Statement declaring this commercial affiliation, as well as a statement regarding the Role of Funders in your study. If the funding organization did not play a role in the study design, data collection and analysis, decision to publish, or preparation of the manuscript and only provided financial support in the form of authors' salaries and/or research materials, please review your statements relating to the author contributions, and ensure you have specifically and accurately indicated the role(s) that these authors had in your study. You can update author roles in the Author Contributions section of the online submission form.

iCARE is not a commercial company. iCARE is an international research study that represents a collaboration between multiple international research teams. The project is led out of the Montreal Behavioural Medicine Centre by Drs. Lavoie and Bacon. As such, iCARE does not have a funding role it is merely the name of an international research study led by the Canadian researchers.

4. Please also include the following statement within your amended Funding Statement. “The funder provided support in the form of salaries for authors [insert relevant initials], but did not have any additional role in the study design, data collection and analysis, decision to publish, or preparation of the manuscript. The specific roles of these authors are articulated in the ‘author contributions’ section.”

None of the funders provided salary support for any of the authors. Instead, funding bodies listed provided operational funds for the research to be conducted. For example, contributing funds to pay research staff that assisted with data collection and participant incentives.

5. Please also provide an updated Competing Interests Statement declaring this commercial affiliation along with any other relevant declarations relating to employment, consultancy, patents, products in development, or marketed products, etc. Within your Competing Interests Statement, please confirm that this commercial affiliation does not alter your adherence to all PLOS ONE policies on sharing data and materials by including the following statement: "This does not alter our adherence to PLOS ONE policies on sharing data and materials.” (as detailed online in our guide for authors http://journals.plos.org/plosone/s/competing-interests) . If this adherence statement is not accurate and there are restrictions on sharing of data and/or materials, please state these. Please note that we cannot proceed with consideration of your article until this information has been declared.

Again, iCARE is not a commercial company. It does not produce profit, nor operate as a business. It is the name of an international research study. As such, there is no violation to PLOS ONE policies on sharing data and materials.

The updated Funding Statement as indicated in query 2 above. As there are no changes to the authors competing interests as iCARE is not a commercial company, nor does it pay any of the co-authors listed here, the Competing Interests Statement remains the same.

7. One of the noted authors is a group or consortium The iCARE Study team. In addition to naming the author group, please list the individual authors and affiliations within this group in the acknowledgments section of your manuscript. Please also indicate clearly a lead author for this group along with a contact email address.

The iCARE project team encompasses staff and trainees who assist with daily research project tasks of the broader iCARE study team. This list changes monthly based on student, trainee, volunteer and staff changes. As such, it is not possible to produce a list of all the individuals involved on the iCARE study team. There is also no lead author for the iCARE study beyond Drs. Lavoie and Bacon, both Drs. Lavoie and Bacon are the leaders of the entire iCARE study team. This text has been added to the acknowledgements section of the manuscript along with the general project email address. We have also added the Montreal Behavioural Medicine Centre (MBMC) affiliation to the iCARE Study team to make the connections clearer to a reader.

8. In the online submission form, you indicated that this is secondary analysis of cross-sectional data collected as a part of the International COVID-19 Awareness and Responses Evaluation (iCARE) Study led by the Montreal Behavioural Medicine Centre (MBMC; a joint Centre intégré universitaire de santé et de services sociaux du Nord-de-l’Île-de-Montréal (CIUSSS- NIM)/Université du Québec à Montréal/Concordia University academic research and training centre). All data available by request from the www.icarestudy.com website. All PLOS journals now require all data underlying the findings described in their manuscript to be freely available to other researchers, either a. In a public repository, b. Within the manuscript itself, or c. Uploaded as supplementary information. This policy applies to all data except where public deposition would breach compliance with the protocol approved by your research ethics board. If your data cannot be made publicly available for ethical or legal reasons (e.g., public availability would compromise patient privacy), please explain your reasons on resubmission and your exemption request will be escalated for approval.

As this is a secondary data analysis, the original ethics for the iCARE project have outlined that the data is stored and administered through the iCARE Study online repository. All data we used can be provided by the iCARE study team through a request to made on the iCARE website indicated. The iCARE Study team manages all project data securely on their servers at the MBMC as a public repository – anyone can request access to the data via the online repository the iCARE Team manages. As these are the procedures in place for the iCARE study itself, we are unable to alter them as we are merely using the freely available data provided by the iCARE Study team. Based on feedback of reviewers 1,2 and 3 in the “Have the authors made all data underlying the findings in their manuscript fully available?” question all responding “yes,” we believe that it is clear to the reader that all data we used can be publicly accessed for free.

9. Please remove all personal information, ensure that the data shared are in accordance with participant consent, and re-upload a fully anonymized data set.

None of the uploaded data we have provided in tables or figures contains personal information. No columns are hidden. If the journal can confirm where they have seen personal information in the data presented, we are more than happy to ensure it is appropriately removed.

Three studies that have previously been published were suggested by reviewer 3. We have carefully reviewed these studies and included one of the suggested ones in the revised manuscript.

Reviewer #1 comments:

1. In the abstract, you should provide full name of COVID-19.

The full name of severe acute respiratory syndrome coronavirus 2 has been added in the abstract and abbreviated as SARS-CoV-2.

2. Please provide full name of OR.

The full name of ‘odds ratio’ has been added.

3. You need to give an example of the tailored program.

An example has been written in text as “public health campaign promoting culturally relevant physical activity in response to greater perceived psychological distress.” We have also specified that we are talking about a tailored behavioural intervention.

4. I just wonder, why you used P<.1 to put variables into the equation and <0.1 for the significant level, normally we used 25%.

We believe the second p-value here above may be a typo. We used p<.10 to explore if an individual independent variable (i.e., psychological distress and infection worry) and sex warranted inclusion in the final model with both independent variables present. This process of conservative checking is one that is used in model building in a variety of settings and is more conservative than using .25 (See example reference below). In the final model with both psychological distress and infection worry we used a p-value <.01 instead of the typical p<.05. In both cases we chose to use more conservative values as we run a large number of models and aimed to reduce the likelihood of spontaneous significance from arising in our findings.

Deslippe AL, González OD, Buckler EJ, Ball GDC, Ho J, Bucholz A, Morrison KM, Mâsse LC. Do Individual Characteristics and Social Support Increase Children's Use of an MHealth Intervention? Findings from the Evaluation of a Behavior Change MHealth App, Aim2Be. Child Obes. 2023 Oct;19(7):435-442. doi: 10.1089/chi.2022.0055. Epub 2022 Dec 22. PMID: 36576875.

Reviewer #2 comments:

1. Please consider revising the title to make it more concise.

We have shortened the title to “How sex impacted associations between psychological distress and worry on adults’ health behaviours during SARS-CoV-2.”

2. The abstract effectively sets the context and highlights key findings of the study, but a few improvements can enhance clarity including: Consider making the abstract short and precise as it is written very long, clarify the implications of sex as a moderator in the results, ensure consistent terminology and keep statements concise for better readability.

We have shortened the text in the abstract from the maximum word count of 300 to 270. We have clarified the meaning of sex in the context of moderation in Columbia (“…with females experiences more delirious changes (p<.01)”) and replaced terminology throughout to repeat the same words.

3. The introduction effectively outlines the impact of COVID-19. However, consider clarifying some points for better flow and coherence. For instance, provide clearer transitions to enhance readability when mentioning studies and ensure that the rationale for the country-level comparison is clearly articulated, emphasizing its relevance to the study's objectives.

We have altered wording throughout the introduction to make it clearer when a previously published study is being discussed and specified the country of each study to emphasize diversity in trends across countries. We have also adapted the wording in lines 818-826 to re-iterate why country comparisons are useful to establish sex-based trends (as opposed the gender-based trends that can impact health behaviours).

4. Specify the criteria for compensation to ensure transparency when describing participant recruitment.

We have altered the wording in lines 870-871to emphasis the individual approach that each polling service in the respective countries employed across the iCARE study.

5. The eligibility criteria could be simplified for better readability.

We have simplified the wording to:

To be included in this analysis, participants from Canada, Columbia and Ireland who completed an online survey from April 2020-December 2022 who reported changes in their health behaviours “since the start of COVID-19 (referring to SARS-CoV-2)” were assessed in this analysis. (Line 886-888)

6. Clarify how the internal consistency of the infection worry variable was assessed

We had used and presented Pearson correlations of each of the four items assessing worry in a pairwise fashion. We used a cut off of .80 to indicate high correlation and inappropriateness of creating a summary worry variable. Correlations were assessed in each country. We have adapted the metric to present Cronbach Alpha’s instead now as this is a more typical and recognized metric that can be used to indicate summary variable reliability (lines 931-933).

7. How are the statistical analysis methods clearly linked to the research questions?

We have revised the Data Analysis section to make the links between what was done and the research question clearer.

8. The results section effectively presents key findings across the three countries, highlighting significant differences in health behaviors related to psychological distress and infection worry. However, consider providing clearer summaries of the statistical findings for each country to enhance readability. For instance, clarify the impact of demographic factors, such as age and clinical diagnoses, on the results to contextualize the findings better.

Given that clarifying the impacts of demographic factors, such as age, on the result to contextualize them beyond present the values themselves (as shown) would enter more into a discussion space we have not incorporate this suggestion in the results section. We have however, articulated the impact of co-variates more clearly in the discussion section (see lines 118

---

## [Decision Letter · Decision Letter 1]

23 Nov 2025

Dear Dr. Cohen,

Thank you for submitting your manuscript to PLOS ONE. After careful consideration, we feel that it has merit but does not fully meet PLOS ONE’s publication criteria as it currently stands. Therefore, we invite you to submit a revised version of the manuscript that addresses the points raised during the review process.

We look forward to receiving your revised manuscript.

Kind regards,

Prof. Anat Gesser-Edelsburg, Ph.D.

Academic Editor

PLOS ONE

Journal Requirements:

Reviewers' comments:

Reviewer's Responses to Questions

**Comments to the Author**

Reviewer #1: All comments have been addressed

Reviewer #3: All comments have been addressed

2. Is the manuscript technically sound, and do the data support the conclusions?

Reviewer #1: Yes

Reviewer #3: Yes

3. Has the statistical analysis been performed appropriately and rigorously?

Reviewer #1: Yes

Reviewer #3: Yes

4. Have the authors made all data underlying the findings in their manuscript fully available?

Reviewer #1: Yes

Reviewer #3: Yes

5. Is the manuscript presented in an intelligible fashion and written in standard English?

Reviewer #1: Yes

Reviewer #3: Yes

Reviewer #1: Thank you for your revision. You have addressed my concern. I am really satisfied with revision. It is a good manuscript for publication.

Reviewer #3: Title: How sex impacted associations between psychological distress and worry on adults’

health behaviours during SARS-CoV-2

MS ID: PONE-D-25-24290R1

General Comments:

After reading over the three reviewers’ comments on the original manuscript, I am impressed with authors efforts, edits, and the changes made to the original manuscript. The revised manuscript is a much better write-up.

My only reservation involves the how the three nations were selected (i.e., Canada, Columbia and Ireland). I wonder if the authors could clarify how these three nations were selected out of the 42 nations that completed (or nearly completed) the electronic surveys.

I note the Gunther Eysenbach citation, and understand the use of CHERRIES. I note that the three nations had good completeness scores (98%, 95.5% and 78.7%). With that being noted, how again were Canada, Columbia and Ireland selected from the original 42 nations? For instance, were these nations selected because they represented the “top three” completeness scores, thus more statistically stable than the other 39 other nations?

Maybe you could include the mean and ranges of the completeness score of all 42 nations (as you would in a basic Method section under a Participant section description of a human participant sample).

For example, (“… the average completeness score of the 42 nations was 50%, ranging from 23% to 98%.”). Again, if I am missing something here, I apologize; but I do not read in your revision, how Canada, Columbia and Ireland were specifically selected out of the 42 nations that had some chance or opportunity to be included in your analysis.

Again, were the three nations selected because they were the top three? Or was another consideration used for inclusion. Please note the difference between stating “Eligibility for this analysis” (line 156) and “Inclusion for this analysis”.

The manuscript notes eligibility as defined as completeness (connection to Eysenbach), but the paper does not mention the overall baseline or a comparison score to the other less eligible nations (the other 39 countries). Of course, this the asks the question, how was inclusion (selection) of a nation/country defined? How was this inclusion unbiased, and unbiased given the completeness scores of the other 39 nations?

Thank you for your revision.

**Do you want your identity to be public for this peer review?** For information about this choice, including consent withdrawal, please see our Privacy Policy

Reviewer #1: **Yes: ** Nitikorn Phoosuwan

Reviewer #3: No

---

## [Author Response · Author response to Decision Letter 2]

24 Nov 2025

We thank the reviewers and editor for their time reviewing our revisions to this manuscript. Below we have outlined all queries and our responses in italics. Unmarked and marked up versions of the revised manuscript have been attached as separate documents in the online portal. Line numbers referenced below correspond to the marked-up version in the portal (not the unmarked version).

Journal feedback:

Three studies that have previously been published were suggested by reviewer 3. We have carefully reviewed these studies and included one of the suggested ones in the revised manuscript.

We have reviewed our reference list and all references are accurate. We have not included any redacted papers in our reference list to the extent of our knowledge.

Reviewer #3 comments:

After reading over the three reviewers’ comments on the original manuscript, I am impressed with authors efforts, edits, and the changes made to the original manuscript. The revised manuscript is a much better write-up.

1. My only reservation involves the how the three nations were selected (i.e., Canada, Columbia and Ireland). I wonder if the authors could clarify how these three nations were selected out of the 42 nations that completed (or nearly completed) the electronic surveys. I note the Gunther Eysenbach citation, and understand the use of CHERRIES. I note that the three nations had good completeness scores (98%, 95.5% and 78.7%). With that being noted, how again were Canada, Columbia and Ireland selected from the original 42 nations? For instance, were these nations selected because they represented the “top three” completeness scores, thus more statistically stable than the other 39 other nations?

Maybe you could include the mean and ranges of the completeness score of all 42 nations (as you would in a basic Method section under a Participant section description of a human participant sample).

For example, (“… the average completeness score of the 42 nations was 50%, ranging from 23% to 98%.”). Again, if I am missing something here, I apologize; but I do not read in your revision, how Canada, Columbia and Ireland were specifically selected out of the 42 nations that had some chance or opportunity to be included in your analysis.

Again, were the three nations selected because they were the top three? Or was another consideration used for inclusion. Please note the difference between stating “Eligibility for this analysis” (line 156) and “Inclusion for this analysis”.

The manuscript notes eligibility as defined as completeness (connection to Eysenbach), but the paper does not mention the overall baseline or a comparison score to the other less eligible nations (the other 39 countries). Of course, this the asks the question, how was inclusion (selection) of a nation/country defined? How was this inclusion unbiased, and unbiased given the completeness scores of the other 39 nations?

Selected of Canada, Ireland and Columbia had nothing to do with their completion rate. We present the completion rate as it is important to report clearly to increase the transparency of data reporting using cross-sectional studies (see the CHERRIES checklist for reference to this criteria).

As we indicated on line 134-136 “Canada, Columbia and Ireland were selected for data analysis as these three countries had complete nationally representative data collected for questions relevant to measures assessing changes in mental well-being and health behaviours attributed to SARS-CoV-2.”

We can appreciate that this may still be confusing as to why the three countries were chosen. First, these three countries had nationally representative data samples recruited based on demographic factors – not all of the other countries that participated in the iCARE study had this. To help minimize responder bias between countries based on a demographic confounder we believe it is important to use nationally representative data to get a clearer picture based on individual country context. Only Australia, Canada, Colombia, France, Ireland, Israel, Italy, the United Kingdom and the United States collected representative data and would fit this.

The second important factor was that these three countries had the same wording in how changes to health behaviours was asked. Not all countries collected the exact same data, and not all countries used the same wording in their surveys. This was done to give researchers of different countries some autonomy to tailor the cross-sectional surveys to respond to information that their own health officials wanted to have collected.

The third important factor is that these three countries specifically had their waves of data collection completed and cleaned. The iCARE team had a few other countries that were actively still recruited data using the same type of questions of nationally representative samples, but these were not finished collection and thus, not available to use yet (e.g., Italy and France). When these countries are fully collected and cleaned it would be useful to consider the methods of this analysis in those countries to see if trends we identified in Canada, Columbia and Ireland prevail. To re-iterate the multi-faceted nature of country selected for our analysis we have expanded the text to include the additional information (Line 133-145):

“Of the 42 countries that participated in the iCARE study, Australia, Canada, Colombia, France, Ireland, Israel, Italy, the United Kingdom and the United States collected data using representative samples of their respective populations based on demographic characteristics. To limit responder bias across countries on any potential confounding demographic factors, we explored data from nationally representative samples in the iCARE project that were fully collected, cleaned and asked about changes in mental well-being and health behaviours attributed to SARS-CoV-2. This left us with three countries that were eligible to use: Canada, Columbia and Ireland.”

We have also added the use of the three countries as a limitation to make it clear that we believe the analyses would have benefited from having included 42 countries – but it was not possible given how the data was collected (line 409-414):

“Finally, we were unable to explore data from all 42 countries in the iCARE Study due to differences in how samples were collected from diverse countries (i.e., nationally representative versus convenience), which questions were asked (i.e., countries could tailor their wording to national health aims) and which countries had their data fully collected and cleaned (i.e., not in progress). In would be incredibly insightful to explore trends across all 42 countries, had it been possible, to consider a more complete narrative of when trends may prevail and why.”

---

## [Decision Letter · Decision Letter 2]

4 Dec 2025

How sex impacted associations between psychological distress and worry on adults’ health behaviours during SARS-CoV-2.

PONE-D-25-24290R2

Dear Dr. Cohen,

We’re pleased to inform you that your manuscript has been judged scientifically suitable for publication and will be formally accepted for publication once it meets all outstanding technical requirements.

Kind regards,

Prof. Anat Gesser-Edelsburg, Ph.D.

Academic Editor

PLOS ONE

Additional Editor Comments (optional):

Reviewers' comments:

Reviewer's Responses to Questions

**Comments to the Author**

Reviewer #3: All comments have been addressed

2. Is the manuscript technically sound, and do the data support the conclusions?

Reviewer #3: Yes

3. Has the statistical analysis been performed appropriately and rigorously?

Reviewer #3: Yes

4. Have the authors made all data underlying the findings in their manuscript fully available?

Reviewer #3: Yes

5. Is the manuscript presented in an intelligible fashion and written in standard English?

Reviewer #3: Yes

Reviewer #3: Title: How sex impacted associations between psychological distress and worry on adults’

health behaviours during SARS-CoV-2

MS ID: PONE-D-25-24290R2

General Comments:

The authors have responded to my comments. My confusion specifically focused on how the three nations were selected (Ireland, Canada, and Columbia out of 42 possible nations). The authors have made clear the selection process. I believe, in part, the confusion rested on the language used. See line 134-136 of Revision 1, “… data analysis as these three countries had ‘COMPLETE nationally representative data collected…” (emphasis added). Earlier authors used the word ‘complete’ to refer to completion rates. Thus, attempting to understand the selection of these three nations without further explanation was a burden. Authors rewrites and edits are a welcome addition to the manuscript.

Authors have added further clarity to the Limitations and Strengths section. This section of edits is also welcomed. However, see lines 407 to 409. The sentence beginning with “In would be incredibly insightful ….). Besides the typo (“In” should read “It”, right?). I think this sentence is unnecessary, is an obvious point, and is an unsubstantiated claim. I would request this sentence be deleted.

Another typo, see line 416, and the word “tool”. I believe you mean to use the word “took” – as in “and given the conservative approach we took…”.

I would check again for typos within the manuscripts, specifically correctly spelled words, but incorrectly used words, like the use of “tool” for “took”.

Thank you for an interesting manuscript and research. I appreciate your efforts during this review process.

**Do you want your identity to be public for this peer review?** For information about this choice, including consent withdrawal, please see our Privacy Policy

Reviewer #3: No

---

## [Editor Report · Acceptance letter]

PONE-D-25-24290R2

PLOS One

Dear Dr. Cohen,

I'm pleased to inform you that your manuscript has been deemed suitable for publication in PLOS One. Congratulations! Your manuscript is now being handed over to our production team.

Kind regards,

on behalf of

Prof. Anat Gesser-Edelsburg

Academic Editor

PLOS One